# Query-based Temporal Fusion with Explicit Motion for 3D Object Detection

**Jinghua Hou**[1][*], **Zhe Liu**[1][*] , **Dingkang Liang**[1] , **Zhikang Zou**[2] , **Xiaoqing Ye**[2] ,
**Xiang Bai**[1][†]

[1]Huazhong University of Science & Technology
[2]Baidu Inc.
`{jhhou,zheliu1994,dkliang,xbai}@hust.edu.cn`
`{zhikangzou001}@gmail.com`
`{yexiaoqing}@baidu.com`

## Abstract

Effectively utilizing temporal information to improve 3D detection performance
is vital for autonomous driving vehicles. Existing methods either conduct tem-
poral fusion based on the dense BEV features or sparse 3D proposal features.
However, the former does not pay more attention to foreground objects, leading
to more computation costs and sub-optimal performance. The latter implements
time-consuming operations to generate sparse 3D proposal features, and the perfor-
mance is limited by the quality of 3D proposals. In this paper, we propose a simple
and effective **Q**uery-based **T**emporal Fusion **Net**work (QTNet). The main idea is
to exploit the object queries in previous frames to enhance the representation of
current object queries by the proposed **M**otion-guided **T**emporal **M**odeling (MTM)
module, which utilizes the spatial position information of object queries along
the temporal dimension to construct their relevance between adjacent frames re-
liably. Experimental results show our proposed QTNet outperforms BEV-based
or proposal-based manners on the nuScenes dataset. Besides, the MTM is a plug-
and-play module, which can be integrated into some advanced LiDAR-only or
multi-modality 3D detectors and even brings new SOTA performance with negli-
gible computation cost and latency on the nuScenes dataset. These experiments
powerfully illustrate the superiority and generalization of our method. The code is
available at `https://github.com/AlmoonYsl/QTNet`.

## 1 Introduction

3D object detection is the fundamental task of autonomous driving, which consumes data from
sensors (e.g., LiDAR, Camera) to localize and recognize objects in the 3D world. Recent studies have
explored utilizing temporal information to improve the detection performance. According to different
manners of temporal fusion, we divide them into two categories, including BEV-based (Figure 1(a))
and proposal-based (Figure 1(b)).

BEV-based methods [16, 49, 55, 43, 57] conduct temporal fusion on the dense bird's-eye-view (BEV)
feature by applying an affine transformation to align BEV features in previous frames to enhance
the current BEV features. However, not all pixel-wise BEV features are beneficial for boosting
the detection performance of foreground objects. To alleviate this problem, the proposal-based
approaches [46, 34, 5] usually adopt a region proposal network (RPN) to obtain many 3D proposals

---

[*]Equal contribution.

[†]Corresponding author.

37th Conference on Neural Information Processing Systems (NeurIPS 2023).

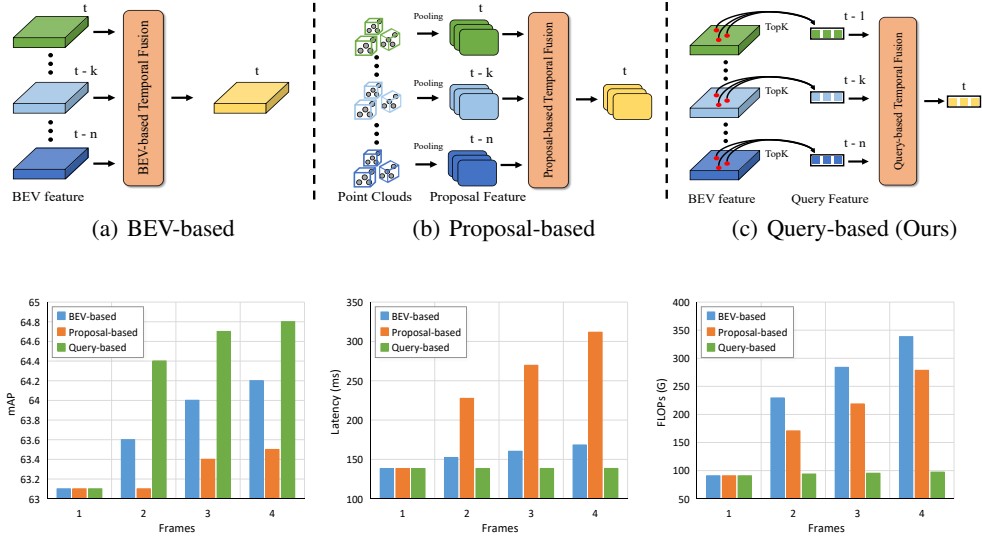

(a) BEV-based  (b) Proposal-based  (c) Query-based (Ours)

(d) Comparison of different paradigms on the same baseline

Figure 1: **Temporal fusion architecture comparison. (a)** BEV-based: conduct temporal fusion on the dense BEV features. **(b)** Proposal-based: first extract proposal by 3D RoI pooling operation and then implement temporal fusion on the proposal-level features. **(c)** Query-based (Ours): directly adopt the query-level features to achieve temporal fusion. **(d)** We compare the performance, computational cost, and latency of different paradigms on the same baseline (TransFusion-L [2]). Computation cost and latency are tested on a single NVIDIA RTX 4090 GPU with 1 batch size. For BEV-based and proposal-based paradigm, we reproduce the representative long-term fusion of MGTANet [16] and MPPNet [5] on the baseline, respectively.

and then achieve temporal fusion by focusing on foreground object features extracted by the time-consuming 3D RoI-like operations [37, 36, 9]. However, the effectiveness of temporal fusion is bounded by the quality of the generated 3D proposals.

We argue that an effective and lightweight temporal fusion module is more meaningful to autonomous driving technology with high real-time and detection accuracy requirements. Towards this goal, we rethink the format of feature representation and the temporal modeling paradigm in temporal fusion for 3D object detection task, and propose a new query-based temporal fusion method called QTNet. Different from the BEV-based and proposal-based feature representation in Figure 1(a) and Figure 1(b), we adopt a query-based mechanism shown in Figure 1(c) that possesses two merits. On the one hand, the query-based representation is sparse and can effectively aggregate the foreground object information by attention operation [39]. On the other hand, the query-based format is more efficient due to getting rid of complex 3D RoI operations and is less sensitive to 3D object of size and orientation than proposal-based representation.

However, the temporal modeling based on query representation is non-trivial to work well. A natural manner is to compute the feature similarity between the queries of the current and previous frame by cross attention operation. However, the feature measurement is less reliable since 3D objects with similar geometric structures between adjacent frames are difficult to distinguish, leading to failed matching. Fortunately, we observe that 3D objects usually follow the physical law of motion in the real-world 3D space, which means that the same 3D object between adjacent frames does not shift too much in a short time.

Based on the above observations, we further adopt a Motion-guided Temporal Modeling (MTM) module based on the sparse query representation. Specifically, we first align objects from previous frame to current frame and then generate the attention map by explicit position information of objects. Finally, we aggregate the temporal features by the attention map to enhance the current query features for the subsequent refinement and produce the final detection results.

Besides, as shown in Figure 1(d), we compare our query-based paradigm with the BEV-based and the proposal-based paradigm in terms of FLOPs, latency, and performance. We can observe that our QTNet possesses distinct advantages in performance, computation cost, and latency. Overall, our contributions are summarized as follows: **(i)** We propose a new temporal fusion method called QTNet for 3D object detection based on sparse query-based feature representation, which is more effective and efficient than BEV-based and proposal-based manners. **(ii)** We propose the MTM, which can be plugged into LiDAR-only or multi-modality 3D detectors and boost their performance with negligible computation cost and latency.

## 2 Related work

**3D object detection**    The current 3D object detection detectors can be roughly categorized into three categories: LiDAR-based, camera-based, and multi-modality. For LiDAR-based, some methods [37, 56, 54, 52] directly consume pnoit clouds and predict the 3d detection results. Other methods [17, 27, 36, 50, 2, 18] first quantify point clouds into regular grid structures (e.g., voxels, pillars) and then utilize a 3D backbone to extract features. These methods map 3D grids to bird-eye view (BEV) representation by height compression and utilize 2D CNN to extract features further for 3D object detection with efficiency. For camera-based, some methods [21, 12, 20, 19] lift the image features into 3D space and then generate BEV features. Other methods [23, 42] follow the DETR [4] paradigm, and they utilize queries to interact with the multi-view image features and predict the 3D detection results. LiDAR-based methods have precise geometric information with high performance but lack dense texture information. In contrast, camera-based methods have rich texture information but lack explicit 3D geometric information. Obviously, LiDAR and camera information are complementary. Therefore, multi-modality methods [14, 24, 51, 8, 2, 28, 22, 47, 15, 26, 25] utilize multi-modal fusion to improve detection performance.

**3D multi-object tracking**    Due to the lack of rich texture information for point clouds, LiDAR-based tracking methods [41, 50, 31, 33] usually utilize the motion of objects for the association. For example, CenterPoint [50] estimates the velocity by an additional regression head and utilizes L2 distance to associate objects in the previous and current frame. SimpleTrack [33] explores settings of tracking on different datasets through a series of experiments. With the development of camera-based detectors, some methods [53, 32] have begun to explore conducting 3D multi-object tracking on the camera domain since there is rich appearance information in images. MUTR3D [53] uses queries to represent objects and achieves tracking by attention [39]. PF-Track [32] utilizes a temporal transformer to conduct temporal fusion and integrates past and future reasoning for tracked objects.

**3D object detection with temporal fusion**    In the real world, temporal information is beneficial to perceive the localization and direction of movement for 3D objects. Towards this goal, some researchers exploit temporal information to boost the performance of 3D detection task. There are two sets of mainstream, including BEV-based [49, 21, 16, 13] and proposal-based [46, 34, 5] temporal fusion. For BEV-based paradigm, the typical approach MGTANet [16] adopts an SM-VFE module to encode LiDAR point features and then a motion-guided deformable module to align and aggregate multi-frame BEV features for temporal fusion. However, this BEV-based paradigm takes both the foreground and background into consideration, leading to additional computation costs and sub-optimal performance. For proposal-based paradigm, MPPNet [5] utilizes RoI grid pooling [9] to extract proposal features and generate trajectories for temporal fusion. However, this proposal-based paradigm usually adopts a multi-stage pipeline and some time-consuming 3D RoI pooling operation, which makes the whole pipeline complex and brings unacceptable runtime. In summary, the above methods have the disadvantage of sub-optimal performance or high latency and computation cost. To address these problems, in this paper, we propose an efficient temporal fusion method QTNet based on query representation, which achieves better performance and notably reduces the computation cost and latency.

## 3 Method

In this section, we introduce the proposed **Q**uery-based **T**emporal Fusion **Net**work (QTNet), which is presented in Figure 2. QTNet consists of a DETR-like 3D detector, a memory bank, and our proposed Motion-guided Temporal Modeling (MTM) module. In contrast to the prior arts, including

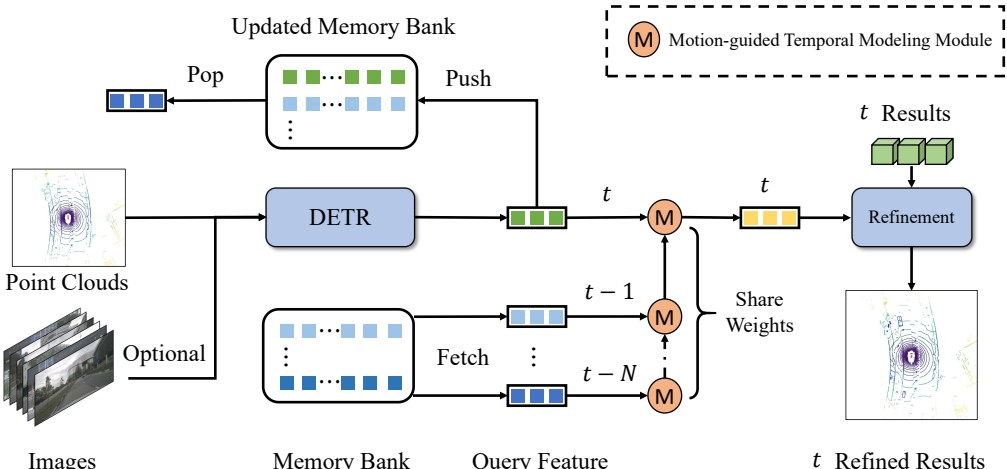

Figure 2: **The overall architecture of the proposed QTNet.** We first feed sensor data to the DETR-like 3D detector and generate current frame queries. We utilize a memory bank to store historical queries for query-based temporal fusion, which follows the FIFO rule. In the temporal fusion, we fetch these historical queries from the memory bank and then feed them to our motion-guided temporal modeling module to fuse queries in different frames. Finally, the fused queries are utilized to refine current detection results.

BEV-based [16, 5, 49, 57] and proposal-based [34, 5], we conduct the temporal fusion on sparse query representation. Specifically, we first feed LiDAR point clouds to DETR-like 3D detectors [2, 47, 15] and generate sparse query features. Note that our method can also support multi-modal data as input (e.g., images and LiDAR point clouds). Thus, the input of images is optional. Then, we reserve the historical sparse query features by a memory bank to reduce repeated computation costs. Next, we take the query features of the previous frame and current frame into our MTM module and produce the enhanced query features for the final 3D detection. Next, we will present the details of the main components in QTNet.

## 3.1 Memory Bank

Different from the existing temporal fusion methods [16, 49, 57] that require much repeated computation for the feature extraction from the previous frames, we utilize a memory bank to preserve the historical information for the subsequent temporal fusion. Here, we define a memory bank $S$ with the size of $N \times K$, where $N$ means the number of used historical frames and $K$ is the number of queries. Our memory bank follows the first-in and first-out (FIFO) rule to achieve query feature updates. In more detail, we first conduct temporal fusion in the $t$ frame and reserve $N$ historical frames. Then, we fetch queries $[Q_{t-1}, Q_{t-2}, ... Q_{t-N}]$ in historical frames. Next, we can fuse these historical queries with the current queries $Q_t$ so as to obtain the enhanced current queries $Q'_t$. Finally, we push current queries $Q_t$ into $S$ and pop the oldest queries $Q_{t-N}$.

## 3.2 Motion-guided Temporal Modeling Module

The Motion-guided Temporal Modeling (MTM) module is applied to enhance the current query features by combining the feature information from the history frames. Specifically, we first obtain the current object queries $Q_t$ and their corresponding positions $C_t$ (can also be regarded as the center of 3D objects) from the DETR-like 3D detectors. In this paper, we adopt Transfusion-L [2] as the default 3D detector. Then, we feed current queries $Q_t$, previous fused queries $Q'_{t-1}$, and their corresponding positions $C_t$ and $C_{t-1}$ to MTM for generating current fused queries $Q'_t$. In more detail, we present the structure of MTM in Figure 3, which includes three steps: the position alignment of objects between the previous and the current frames, the attention map generation, and the feature aggregation. For the process of this alignment, we first align objects from previous frame to current frame by the motion of objects and the ego vehicle. Given the ego pose matrix $R_w^{t-1}$ and $R_w^t$ in $t-1$

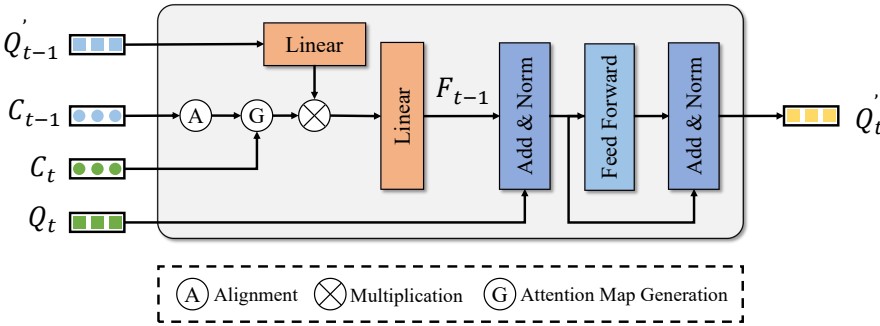

Figure 3: **The details of MTM.** In the MTM, the center of queries in different frames is used to produce the attention map of queries, which is then applied to enhance the current queries.

frame and $t$ frame, we transform objects from the world coordinate to the ego coordinate. Then, the aligned transformation matrix $R_{t-1}^t$, which transforms objects from $t-1$ frame to $t$ frame, can be calculated as:

$$R_{t-1}^t = R_w^t \cdot \mathrm{inv}(R_w^{t-1}), \tag{1}$$

where the $\mathrm{inv}$ denotes the matrix inversion operation. Then, we can align the center of objects in $t-1$ frame to $t$ frame by their velocity $V_{t-1}$ and the aligned transformation matrix $R_{t-1}^t$ with the related time interval $\Delta t$ between $t$ and $t-1$ frame:

$$C_{t-1}' = (C_{t-1} + V_{t-1} \cdot \Delta t) \cdot (R_{t-1}^t)^T \tag{2}$$

For the attention map generation, we first define the number of $Q_t$ is $N_t$ and $Q_{t-1}$ is $N_{t-1}$. Then, we utilize the L2 center distance of objects between adjacent frames to build the cost matrix $O_{t-1}^t \in \mathbb{R}^{N_t \times N_{t-1}}$:

$$O_{t-1}^t = \mathrm{L}_2(C_t - C_{t-1}'), \tag{3}$$

where the $\mathrm{L}_2$ denotes the euclidean distance. To avoid the occurrence of impossible matching situations for better temporal modeling, we introduce the L2 velocity error distribution of each category in $0.5$ seconds as our distance cost threshold $\gamma$, which is similar to CenterPoint [50]. Besides, these associated objects with different categories are masked. We formulate the above process as follows:

$$M = \begin{cases} 0, & O_{t-1}^t \le \gamma \quad and \quad s_t = s_{t-1} \\ 1e^8, & O_{t-1}^t > \gamma \quad or \quad s_t \ne s_{t-1} \end{cases}, \tag{4}$$

where $s_t$ and $s_{t-1}$ denote the category of objects in $t$ and $t-1$ frames, respectively. After adopting the attention mask $M$ with the row-wise softmax function on the cost matrix, we generate the attention map $A$, which is derived from the explicit position information:

$$A = \mathrm{softmax}(-1 \cdot O_{t-1}^t - M), \tag{5}$$

For feature aggregation, we utilize the attention map to aggregate historical features. Given the current queries $Q_t$, previous fused queries $Q_{t-1}'$, and attention map $A$ in Figure 3, the aggregated temporal features $F_{t-1}$ can be calculated as:

$$F_{t-1} = \phi_2(A \cdot \phi_1(Q_{t-1}')), \tag{6}$$

where the $\phi_1$ and $\phi_2$ denote two linear layers. After aggregation, we can combine $F_{t-1}$ and $Q_t$ to generate the current fused queries $Q_t'$:

$$Q_t' = \mathrm{Norm}(Q_t + \mathrm{Dropout}(\mathrm{FFN}(\mathrm{Norm}(Q_t + \mathrm{Dropout}(F_{t-1}))))), \tag{7}$$

where Dropout, Norm, and FFN denote the dropout operation [38], the layer normalization [1], and the feed-forward network, respectively. Finally, we utilize $Q_t'$ to refine the predicted results of the current frame through two simple FFNs, which are utilized to estimate the confidence of the classification branch and the residual of the regression branch for further refinement. Besides, we follow [40, 35] to utilize the IoU branch to rectify the confidence score of prediction results in the current frame.

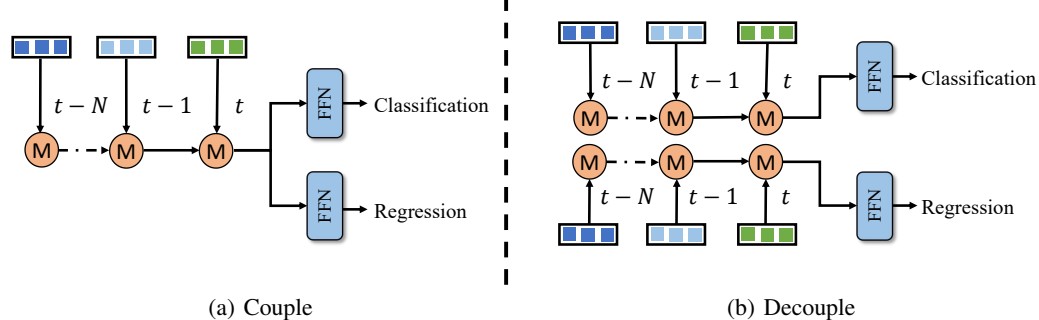

(a) Couple          (b) Decouple

Figure 4: **Decouple temporal fusion.** The decoupling manner conducts temporal fusion on classification and regression branches separately for better detection performance. Here, the weights of classification and regression branches are not shared.

### 3.3 Decouple Strategy

We observe that there is an imbalance in the classification and regression learning for temporal fusion. This phenomenon is also mentioned in VISTA [10] for 3D detection task. However, this decoupled manner brings more computation costs and latency, which is unacceptable for most BEV-based and proposal-based methods. In contrast, benefiting from our efficient design of temporal fusion, the computation cost and latency of QTNet are negligible compared with the whole 3D detection network. Therefore, we decouple the classification and regression branches as shown in Figure 4(b) instead of the coupling manner in Figure 4(a) that is usually employed in the previous temporal fusion methods [49, 5, 16]. Specifically, we separately implement the temporal fusion for the classification branch and the regression branch so as to achieve better detection performance.

## 4    Experiments

### 4.1    Experimental setup

**Dataset**    We evaluate our method on the nuScenes dataset [3], which is a large-scale autonomous driving dataset. It contains 700, 150, and 150 scenes for training, validation, and testing. Each scene is roughly 20 seconds long and annotated at 2Hz, and provides point clouds acquired by 32-beam LiDAR and surrounding images acquired by 6 cameras. Note that each sample consists of one key frame and nine sweep frames. Besides, there are roughly 1.6M objects for 10 categories (barrier, bicycle, bus, car, motorcycle, pedestrian, trailer, truck, construction vehicle, and traffic cone) on the 3D object detection task. For the evaluation metric, Mean Average Precision (mAP) [11] and nuScenes detection score (NDS) [3] are used to evaluate the performance of 3D detectors on the nuScenes dataset.

### 4.2    Implementation details

Following the settings of nuScenes dataset, we concatenate point clouds of one key frame and nine sweeps as the input of a sample, which is roughly 0.5 seconds long. Each point is represented as $(x, y, z, r, \Delta t)$, where the $(x, y, z)$, $r$, and $\Delta t$ denote the 3D coordinate of a point, the reflectance, and the time offset from the key frame, respectively.

Our method is implemented on 3D object detectors with LiDAR or multi-modality (LiDAR and images). For LiDAR, we take the TransFusion-L [2] as our LiDAR-only baseline. For multi-modality, we take the TransFusion [2] and DeepInteraction [47] as our multi-modality baseline. The voxelization range is set to $[-54m, 54m]$ for both $X$ and $Y$ axes and $[-5m, 3m]$ for $Z$ axis. The voxel size is set to $(0.075m, 0.075m, 0.1m)$. We set the number of queries as 200 for training and testing. For temporal fusion, we utilize 2 or 3 historical frames. Note that we do not adopt any test time augmentation (TTA), multiple models ensemble, or future frames during inference.

Table 1: Comparison with state-of-the-art methods on the nuScenes validation set. The L and C represent LiDAR and camera, respectively. The column of Frames denotes the number of key frame. * denotes our reproduced results. † denotes future information is used.

| Method | Year | Modality | Frames | Backbones | mAP↑ | NDS↑ |
|---|---|---|---|---|---|---|
| MVP [51] | NeurIPS 2021 | LC | 1 | DLA34 & VoxelNet | 67.1 | 70.8 |
| AutoAlignV2 [8] | ECCV 2022 | LC | 1 | CSPNet & VoxelNet | 67.1 | 71.2 |
| TransFusion [2] | CVPR 2022 | LC | 1 | R50 & VoxelNet | 67.5 | 71.3 |
| BEVFusion [22] | NeurIPS 2022 | LC | 1 | Swin-Tiny & VoxelNet | 67.9 | 71.0 |
| DeepInteraction [47] | NeurIPS 2022 | LC | 1 | R50 & VoxelNet | 69.9 | 72.6 |
| BEVFusion [28] | ICRA 2023 | LC | 1 | Swin-Tiny & VoxelNet | 68.5 | 71.4 |
| MSMDFusion [15] | CVPR 2023 | LC | 1 | R50 & VoxelNet | 69.3 | 72.1 |
| CMT [44] | ICCV 2023 | LC | 1 | V2-99 & VoxelNet | 70.3 | 72.9 |
| TransFusion* [2] | CVPR 2022 | LC | 1 | R50 & VoxelNet | 67.1 | 70.7 |
| +QTNet | - | LC | 4 | R50 & VoxelNet | 68.5 | 71.6 |
| DeepInteraction* [47] | NeurIPS 2022 | LC | 1 | R50 & VoxelNet | 69.9 | 72.6 |
| +QTNet | - | LC | 4 | R50 & VoxelNet | **70.3** | **73.1** |
| CenterPoint [50] | CVPR 2021 | L | 1 | VoxelNet | 59.6 | 66.8 |
| TransFusion-L [2] | CVPR 2022 | L | 1 | VoxelNet | 65.1 | 70.1 |
| INT [43] | ECCV 2022 | L | 10 | VoxelNet | 61.8 | 67.3 |
| LidarMultiNet [48] | AAAI 2023 | L | 1 | VoxelNet | 63.8 | 69.5 |
| VoxelNeXt [7] | CVPR 2023 | L | 1 | VoxelNet | 60.0 | 67.1 |
| LargeKernel3D [6] | CVPR 2023 | L | 1 | Voxel-LargeKernel3D | 63.3 | 69.1 |
| LinK [30] | CVPR 2023 | L | 1 | Voxel-LinK | 63.6 | 69.5 |
| MGTANet [16] | AAAI 2023 | L | 3 | VoxelNet | 62.9 | 68.7 |
| MGTANet† [16] | AAAI 2023 | L | 3 | VoxelNet | 64.8 | 70.6 |
| TransFusion-L* [2] | CVPR 2022 | L | 1 | VoxelNet | 65.0 | 70.0 |
| +QTNet | - | L | 3 | VoxelNet | **66.3** | **70.8** |
| +QTNet | - | L | 4 | VoxelNet | **66.5** | **70.9** |

Table 2: Comparison with state-of-the-art methods on the nuScenes test set. † denotes future information is used.

| Method | Year | Modality | Frames | Backbones | mAP↑ | NDS↑ |
|---|---|---|---|---|---|---|
| CenterPoint [50] | CVPR 2021 | L | 1 | VoxelNet | 60.3 | 67.3 |
| TransFusion-L [2] | CVPR 2022 | L | 1 | VoxelNet | 65.5 | 70.2 |
| VISTA [10] | CVPR 2022 | L | 1 | VoxelNet | 63.7 | 70.4 |
| LidarMultiNet [48] | AAAI 2023 | L | 1 | VoxelNet | 67.0 | 71.6 |
| VoxelNeXt [7] | CVPR 2023 | L | 1 | VoxelNet | 64.5 | 70.0 |
| LargeKernel3D [6] | CVPR 2023 | L | 1 | Voxel-LargeKernel3D | 65.3 | 70.5 |
| LinK [30] | CVPR 2023 | L | 1 | Voxel-LinK | 66.3 | 71.0 |
| 3DVID† [49] | TPAMI 2021 | L | 3 | VoxelNet | 65.4 | 71.4 |
| MGTANet† [16] | AAAI 2023 | L | 3 | VoxelNet | 65.4 | 71.2 |
| QTNet | - | L | 3 | VoxelNet | 68.2 | 72.0 |
| QTNet | - | L | 4 | VoxelNet | **68.4** | **72.2** |

Our training process is divided into two stages. In the first stage, we train the DETR-like 3D detectors (TransFusion-L, TransFusion, and DeepInteraction) with their default settings. In the second stage, we generate the memory bank of query features and prediction results. Then, we train our QTNet for 10 epochs without GT-Sampling [45] and CBGS [58] on four NVIDIA RTX 4090 GPUs. To optimize our network, we adopt the AdamW [29] optimizer with a one-cycle learning rate policy, and the batch size is set to 16.

## 4.3 Comparison to the state of the art

We compare the proposed QTNet with previous state-of-the-art 3D detectors on the nuScenes [3] validation and test set. As shown in Table 1, QTNet shows consistent performance improvement when our module is integrated into the LiDAR-only detector Transfusion-L [2] or multi-modality methods Transfusion [2] and DeepInteraction [47]. Compared with the single frame LiDAR-only

Table 3: Comparison of computation cost and latency on the nuScenes validation set. * denotes our reproduced results. The FLOPs and Latency are tested on a single NVIDIA RTX 4090 GPU with the batch size of 1.

| Method | Modality | mAP (%)↑ | NDS (%)↑ | FLOPs (G)↓ | Latency (ms)↓ | Params (M) |
|---|---|---|---|---|---|---|
| TransFusion* [2] | LC | 67.1 | 70.7 | 445.9 | 201.2 | 37.0 |
| +QTNet | LC | 68.5 | 71.6 | 446.4 | 207.7 | 37.7 |
| DeepInteraction* [47] | LC | 69.9 | 72.6 | 499.2 | 355.0 | 57.8 |
| +QTNet | LC | **70.3** | **73.1** | 499.7 | 361.5 | 58.5 |
| TransFusion-L* [2] | L | 65.0 | 70.0 | 90.7 | 138.2 | 8.3 |
| +QTNet | L | **66.5** | **70.9** | 90.8 | 144.7 | 8.6 |

Table 4: Compared with other paradigms on the TransFusion-L. * denotes the fine-tuned results. Note that when introducing MGTANet into TransFusion-L, we need to fine-tune the detection head of TransFusion-L, which is harmful to the final detection performance. Therefore, for a fair comparison, we choose the same fine-tuned TransFusion-L on different paradigms.

| Method | Paradigm | mAP | NDS | FLOPs (G) | Latency (ms) | Params (M) |
|---|---|---|---|---|---|---|
| TransFusion-L* [2] | - | 63.1 | 67.8 | 90.7 | 138.2 | 8.3 |
| +MGTANet [16] | BEV | 64.0 (+0.9) | 68.1 (+0.3) | +193.0 | +22.1 | +5.5 |
| +MPPNet [5] | Proposal | 63.4 (+0.3) | 68.2 (+0.4) | +131.3 | +127.9 | +7.0 |
| +QTNet | Query | 64.7 (+1.6) | 69.0 (+1.2) | +0.1 | +4.5 | +0.3 |

baseline TransFusion-L, QTNet brings 1.3% (or 1.5%) mAP and 0.8% (or 0.9%) NDS improvements with 3 (or 4) frames, which achieves a new SOTA performance for LiDAR-only detectors. Besides, QTNet even outperforms MGTANet†, a manner of using the future frame information, by 1.5% mAP and 0.2% NDS. Note that QTNet does not utilize any future information since future information can not be fetched in the realistic scene of autonomous driving. Besides, compared with the single frame multi-modality baseline TransFusion, QTNet also produces 1.4% mAP and 0.9% NDS improvement. Based on the previous SOTA multi-modality method DeepInteraction, QTNet brings 0.4% mAP and 0.5% NDS improvement, achieving a new SOTA result for multi-modality 3D object detection.

On the nuScenes test benchmark, we take TransFusion-L as our baseline. As shown in Table 2, QTNet outperforms the advanced temporal fusion method MGTANet† by 2.8% mAP and 0.8% NDS for 3 frames as input, which illustrates the superiority of QTNet. QTNet achieves 68.4% mAP and 72.2% NDS with 4 frames, setting a new SOTA performance. Besides, we present the computation cost and latency by integrating our QTNet into different baselines in Table 3. It can be seen that our method brings consistent performance improvement with negligible computation cost and latency.

### 4.4 Ablation studies

**Query-based v.s. Other paradigms**    For a fair comparison, we integrate the representative long-term fusion of MGTANet[16] and MPPNet[5] into our baseline TransFusion-L[2] and keep the same number of frames as 3. As shown in Table 4, we present the results of QTNet and other methods in terms of the performance and computation cost. QTNet outperforms the BEV-based method MGTANet and the proposal-based method MPPNet 0.9% NDS and 0.8% NDS, respectively. For computation cost and latency, QTNet only brings a tiny increment of 0.1 GFLOPs and 4.5 ms based on the baseline of TransFusion-L. However, MGTANet even additionally produces 193.0 GFLOPs and 22.1 ms, and MPPNet brings 131.3 GFLOPs and 127.9 ms on the same baseline. These experiments effectively demonstrate that QTNet possesses superior performance and less computation cost and latency when compared with other paradigms. Besides, QTNet is lightweight, which only brings tiny extra parameters of 0.3 M.

**Ablation studies on our QTNet**    We adopt the TransFusion-L [2] as our baseline (I) under the case of 3 frames as input and conduct ablation studies on our QTNet as shown in Table 5(a). Here, we provide an available manner to model the relationship of adjacent frames for temporal fusion by the cross attention operation [39] (II) for comparison. It can be observed that there is only a minor performance improvement with the mAP of 0.2% over baseline. In contrast, our proposed MTM

Table 5: Ablation studies on the temporal fusion, evaluated on the nuScenes validation set.

(a) Ablation studies on components of QTNet.

| # | Cross Attention | MTM | IoU | Decouple | mAP | NDS |
|---|---|---|---|---|---|---|
| I | | | | | 65.0 | 70.0 |
| II | ✓ | | | | 65.2 | 70.0 |
| III | | ✓ | | | 66.2 | 70.5 |
| IV | | ✓ | ✓ | | 66.3 | 70.6 |
| V | | ✓ | ✓ | ✓ | **66.4** | **70.8** |

(b) Ablation studies on the number of frames

| Frames | mAP | NDS | FLOPs (G) | Latency (ms) |
|---|---|---|---|---|
| 1 | 65.0 | 70.0 | 90.65 | 138.2 |
| 2 | 66.1 | 70.6 | +0.07 | +3.1 |
| 3 | 66.4 | 70.8 | +0.10 | +4.5 |
| 4 | **66.5** | **70.9** | +0.14 | +6.5 |
| 5 | 66.4 | **70.9** | +0.24 | +7.7 |

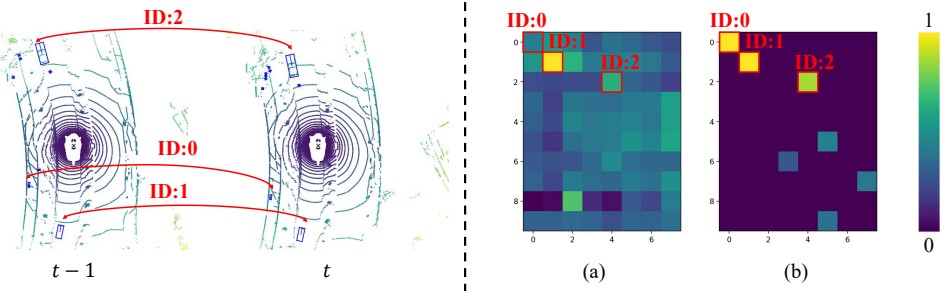

Figure 5: **Attention map comparison.** For better visualization, we select some predictions with high confidence score and calculate the attention map. The row and column of attention map represent objects in $t$ and $t − 1$ frame, respectively. The value of attention map is higher and the relevance is stronger. The attention map (a) generated by cross attention is ambiguous. The attention map (b) generated by our MTM is more discriminative than cross attention.

(III) obtains 1.2% mAP and 0.5% NDS performance gains over baseline. Besides, we apply the IoU branch [35, 40] to rectify the confidence score of prediction results (IV), which can further boost the detection performance. Finally, we decouple the classification and regression of temporal fusion (V), which further brings 0.2% NDS performance gains. The above experiments demonstrate the effectiveness of our proposed components in our QTNet.

**Different Number of Frames** We explore the impact of detection performance on the different number of frames in the temporal fusion, whose results are shown in Table 5(b). We can find that as the number of frames increases from 1 to 4, the performance gain is gradually strengthening. When the number of frames reaches 5, the performance starts to saturate. Besides, the computation cost and latency slightly increase as the number of frames increases. However, thanks to our efficient design, the extra computation cost and latency are much slight compared with our whole network.

**Why MTM is better?** Intuitively, the cross attention operation [39] can directly construct the relevance among query features. However, objects with similar geometric structures between adjacent frames are difficult to distinguish in LiDAR point clouds, which makes the construction of the relationship between two frames less reliable. On the contrary, MTM builds an explicit motion modeling for temporal fusion by taking advantage of the fact that the same 3D objects between adjacent frames do not shift too much in a short time. To further illustrate the effectiveness of our MTM, we present the visualization for the manners of cross attention and our MTM as shown in Figure 5. Here, for convenience, we select three objects from predictions with high scores in $t$ and $t−1$ frames. The attention map generated by cross attention is ambiguous, which leads to sub-optimal performance for temporal fusion. On the contrary, the attention map generated by our MTM is more discriminative. Therefore, our MTM can establish more reliable relevance among queries than cross attention, which proves the rationality and superiority of our proposed MTM.

## 4.5 Limitation

Our method can be integrated into some advanced LiDAR-only or multi-modality DETR-like 3D detectors and brings performance improvements with negligible computation cost and latency. However, the main limitation is that the cost and latency of our method increase as the number of frames

increases. In the future, we will explore recurrent architecture for 3D object detection with temporal fusion.

## 5    Conclusion

In this paper, we propose a query-based temporal fusion network named QTNet for 3D object detection. Based on our query-based feature representation, we introduce a lightweight motion-guided temporal modeling MTM module to aggregate temporal features more efficiently and effectively. Extensive experiments demonstrate the superiority of our proposed QTNet by comparing it with the mainstream BEV-based and proposal-based paradigms. Besides, QTNet even achieves new state-of-the-art on the nuScenes dataset for LiDAR-only and multi-modality 3D detectors with negligible computation cost and latency compared with the baseline network.

**Acknowledgement**    This work was supported in part by the National Science Fund for Distinguished Young Scholars of China (Grant No. 62225603), in part by the Hubei Key R&D Program (Grant No. 2022BAA078), and in part by the Taihu Lake Innovation Fund for Future Technology (HUST: 2023-A-1).

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

# A Appendix

## A.1 Performance on the Camera-based method

Although we design our motion-guided temporal modeling (MTM) module based on the LiDAR domain, we also explore the performance of MTM on camera-based methods. Thus, we integrate the MTM into the advanced camera-based detector CAPE [42] with two frames as input for temporal fusion on the nuScenes [3] validation set. As shown in Table 6, our MTM can also boost the performance of the camera-based method, which effectively demonstrates the generality of our method.

Table 6: Performance of camera-based method with MTM. The C represents camera. * denotes our reproduced results. All models are trained by four NVIDIA RTX 4090 GPUs with 24 epochs and without CBGS [58]. The batch size is set to 4.

| Method | Year | Modality | Frames | Resolution | Backbone | mAP | NDS | mATE | mASE | mAOE | mAVE | mAAE |
|---|---|---|---|---|---|---|---|---|---|---|---|---|
| CAPE* [42] | CVPR 2023 | C | 1 | 704 × 256 | R50 | 27.5 | 35.9 | 0.794 | 0.286 | 0.642 | 0.847 | 0.215 |
| +MTM | - | C | 2 | 704 × 256 | R50 | **31.6** | **43.8** | **0.752** | **0.277** | **0.558** | **0.438** | **0.182** |
| CAPE* [42] | CVPR 2023 | C | 1 | 800 × 320 | V2-99 | 39.7 | 46.3 | 0.693 | 0.270 | 0.438 | 0.747 | 0.206 |
| +MTM | - | C | 2 | 800 × 320 | V2-99 | **43.9** | **53.6** | **0.656** | **0.266** | **0.380** | **0.350** | **0.183** |

## A.2 Performance breakdown for each category

We report the detailed performance of QTNet for each category on the nuScenes [3] testing benchmark, as shown in Table 7. Compared with our LiDAR-only baseline TransFusion-L [2], QTNet brings consistent improvements on most categories, especially on the construction vehicle (+7.0% AP), motorcycle (+6.6% AP), and bicycle (+6.3% AP).

Table 7: Comparison with state-of-the-art methods on the nuScenes testing set for each category. The L and C represent LiDAR and camera, respectively. C.V., Ped., M.C., B.C., T.C., and B.R. represent construction vehicle, pedestrian, motorcycle, bicycle, traffic cone, and barrier, respectively. The column of Frames denotes the number of key frame. † denotes future information is used.

| Method | Modality | Frames | mAP | NDS | Car | Truck | Bus | Trailer | C.V. | Ped. | M.C. | B.C. | T.C. | B.R. |
|---|---|---|---|---|---|---|---|---|---|---|---|---|---|---|
| CenterPoint [50] | L | 1 | 60.3 | 67.3 | 85.2 | 53.5 | 63.6 | 56.0 | 20.0 | 84.6 | 59.5 | 30.7 | 78.4 | 71.1 |
| TransFusion-L [2] | L | 1 | 65.5 | 70.2 | 86.2 | 56.7 | 66.3 | 58.8 | 28.2 | 86.1 | 68.3 | 44.2 | 82.0 | **78.2** |
| VISTA [10] | L | 1 | 63.7 | 70.4 | 84.7 | 54.2 | 64.0 | 55.0 | 29.1 | 83.6 | 71.0 | 45.2 | 78.6 | 71.8 |
| LidarMultiNet [48] | L | 1 | 67.0 | 71.6 | 86.9 | 57.4 | 64.7 | 61.0 | 31.5 | 87.2 | **75.3** | 47.6 | **85.1** | 73.5 |
| VoxelNeXt [7] | L | 1 | 64.5 | 70.0 | 84.6 | 53.0 | 64.7 | 55.8 | 28.7 | 85.8 | 73.2 | 45.7 | 79.0 | 74.6 |
| LargeKernel3D [6] | L | 1 | 65.3 | 70.5 | 85.9 | 55.3 | 66.2 | 60.2 | 26.8 | 85.6 | 72.5 | 46.6 | 80.0 | 74.3 |
| LinK [30] | L | 1 | 66.3 | 71.0 | 86.1 | 55.7 | 65.7 | 62.1 | 30.9 | 85.8 | 73.5 | 47.5 | 80.4 | 75.1 |
| 3DVID† [49] | L | 3 | 65.4 | 71.4 | 87.5 | 56.9 | 63.5 | 60.2 | 32.1 | 82.1 | 74.6 | 45.9 | 78.8 | 69.3 |
| MGTANet† [16] | L | 3 | 65.4 | 71.2 | **87.7** | 56.9 | 64.6 | 59.0 | 28.5 | 86.4 | 72.7 | 47.9 | 83.8 | 65.9 |
| QTNet | L | 3 | 68.2 | 72.0 | 86.5 | 57.2 | **68.3** | 63.0 | 34.3 | 88.1 | 74.9 | 49.7 | 82.7 | 77.0 |
| QTNet | L | 4 | **68.4** | **72.2** | 86.6 | **57.7** | 68.3 | 62.9 | **35.2** | **88.2** | 74.9 | **50.5** | 82.8 | 77.3 |

Besides, we report the detailed performance of QTNet for each category on the nuScenes [3] validation benchmark, as shown in Table 8.

Table 8: Comparison with different baselines on the nuScenes validation set for each category. * denotes our reproduced results.

| Method | Modality | Frames | mAP | NDS | Car | Truck | Bus | Trailer | C.V. | Ped. | M.C. | B.C. | T.C. | B.R. |
|---|---|---|---|---|---|---|---|---|---|---|---|---|---|---|
| TransFusion* [2] | LC | 1 | 67.1 | 70.7 | 87.7 | 61.6 | 75.9 | 42.4 | 26.5 | 88.0 | 75.2 | 63.8 | 77.3 | 72.2 |
| +QTNet | LC | 4 | 68.5 | 71.6 | 87.8 | 63.0 | 76.6 | 43.1 | 27.7 | 89.3 | 77.5 | 68.8 | 78.3 | 72.5 |
| DeepInteraction* [47] | LC | 1 | 69.9 | 72.6 | **88.5** | 64.4 | **79.2** | 44.5 | **30.1** | 88.9 | 79.0 | 67.8 | **80.0** | **76.4** |
| +QTNet | LC | 4 | **70.3** | **73.1** | 88.4 | **64.7** | 79.0 | **44.8** | 29.4 | **89.4** | 80.5 | **70.6** | 79.7 | 76.1 |
| BEVFusion* [28] | LC | 1 | 69.9 | 72.6 | **88.5** | 64.4 | **79.2** | 44.5 | **30.1** | 88.9 | 79.0 | 67.8 | **80.0** | **76.4** |
| +QTNet | LC | 4 | **70.3** | **73.1** | 88.4 | **64.7** | 79.0 | **44.8** | 29.4 | **89.4** | 80.5 | **70.6** | 79.7 | 76.1 |
| TransFusion-L* [2] | L | 1 | 65.0 | 70.0 | 86.7 | 60.4 | 75.3 | 41.6 | 24.6 | 86.8 | 71.8 | 56.5 | 74.4 | **71.8** |
| +QTNet | L | 3 | 66.3 | 70.8 | 87.1 | 61.1 | 75.5 | **43.0** | **25.7** | 87.8 | 75.2 | 61.2 | **75.6** | 71.4 |
| +QTNet | L | 4 | **66.5** | **70.9** | 87.2 | 61.5 | 75.8 | 43.0 | 25.7 | 87.8 | 75.5 | 61.5 | 75.4 | 71.4 |

### A.3 Visualization

To illustrate the superiority of our QTNet, we visualize the results of TransFusion-L [2] on the nuScenes [3] validation set for comparison. As shown in Figure 6, QTNet can detect the hard-detected objects for TransFusion-L and boost the detection performance thanks to our proposed temporal fusion module MTM. As shown in Figure 7, QTNet successfully correct the angle error of objects for TransFusion-L thanks to our proposed temporal fusion module MTM. Besides, as shown in Figure 8, we compare TransFusion-L and QTNet along the temporal dimension for better presentation. It can be seen that the object on the lower left, which is moving away from the ego vehicle, is not detected in $t$ frame by TransFusion-L. However, QTNet can still capture the object in $t$ frame, benefiting from our effective temporal fusion.

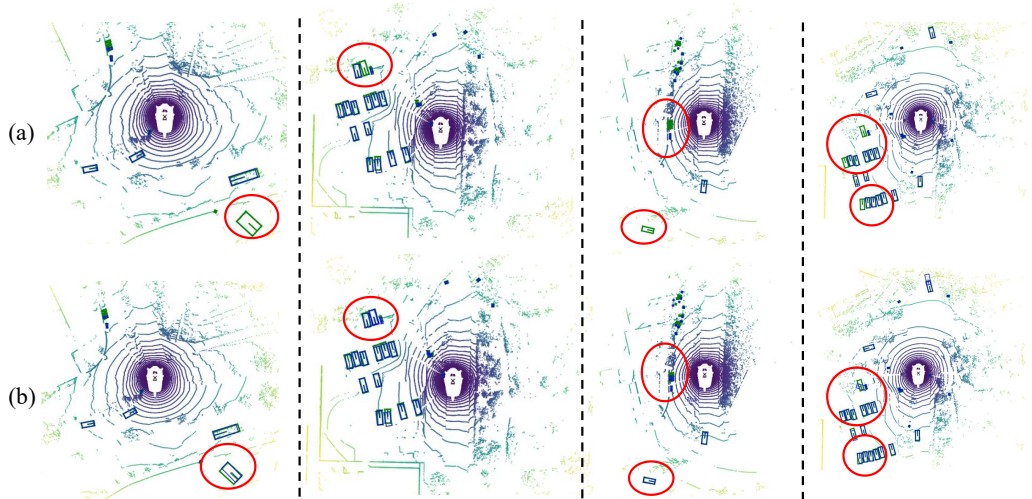

Figure 6: Comparison of LiDAR-only baseline TransFusion-L (a) and QTNet (b) on the nuScenes validation set. Blue and green boxes are the prediction and ground truth boxes. It can be seen that TransFusion-L fails to detect the hard-detected objects. However, thanks to the temporal information, QTNet detects these objects successfully.

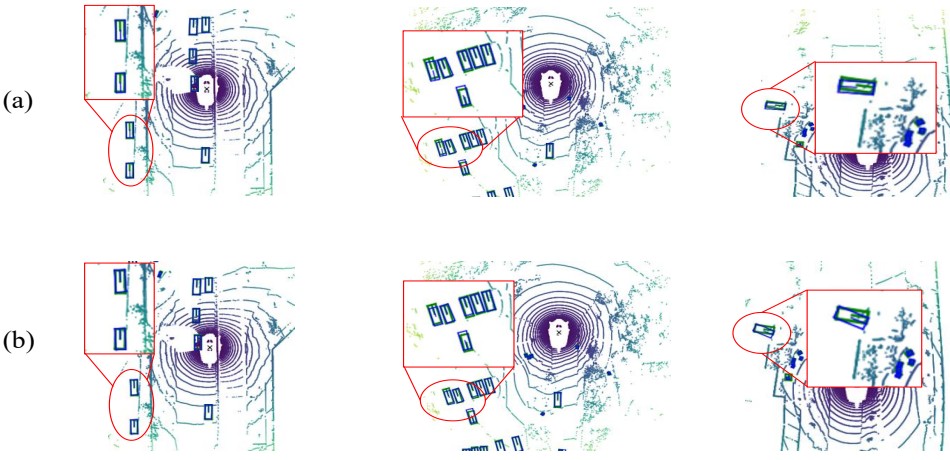

Figure 7: Comparison of LiDAR-only baseline TransFusion-L (a) and QTNet (b) about orientation of objects. Thanks to the temporal information, QTNet successfully corrected the orientation error.

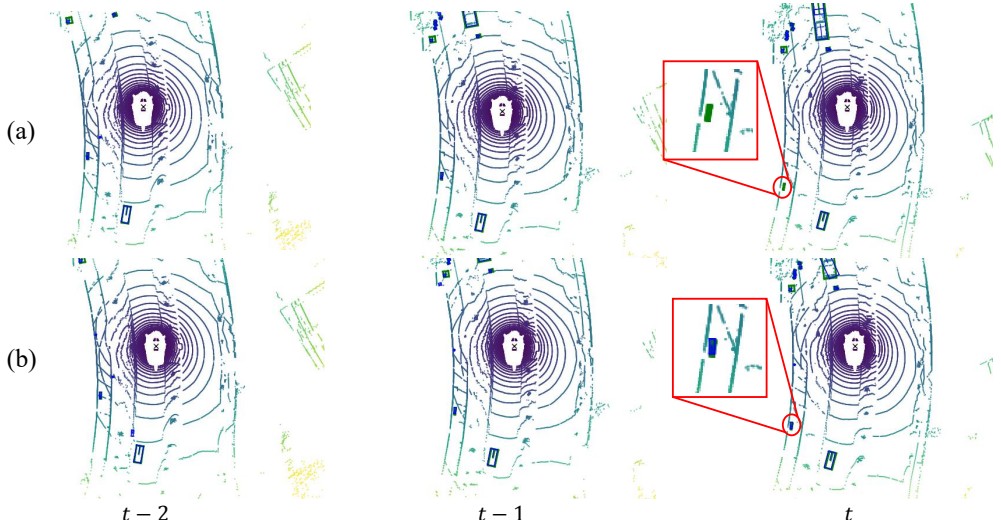

$t-2$ $\quad\quad\quad\quad\quad\quad$ $t-1$ $\quad\quad\quad\quad\quad\quad$ $t$

Figure 8: Comparison of LiDAR-only baseline TransFusion-L (a) and QTNet (b) along the temporal dimension. The ego vehicle is moving from bottom to top.

### A.4  Discussions of potential societal impacts

Effectively utilizing temporal information is vital for autonomous driving. QTNet improves 3D detection performance with negligible computation cost and latency by a lightweight temporal fusion module MTM, which can utilize temporal information to improve the safety of autonomous driving in the real world. However, temporal fusion usually requires sensor synchronization in time, which puts forward higher requirements for the hardware of autonomous driving.

