# OpenReview forum: "Query-based Temporal Fusion with Explicit Motion for 3D Object Detection"
_NeurIPS.cc/2023/Conference — NeurIPS 2023 poster_

### Official Review · Reviewer_LNvD · 2023-06-19

**Soundness:** 2 fair
**Presentation:** 2 fair
**Contribution:** 4 excellent
**Rating:** 4
**Confidence:** 4

**Summary:**

This paper proposes a small component that can be added on-top of single-frame 3D object detectors and perform (late) temporal fusion. This comes at a tiny computational cost and provides some performance improvements.

**Strengths:**

S1) The background, as described in this paper in the introduction and related work sections, is apt yet concise.

S2) The authors identify and tackle an important problem.

S3) The idea -- to aggregate queries over time -- should be appreciated for its simplicity and is different from the approach taken by many prior works, e.g., BEVFormer.

S4) The proposed approach can be added on-top of prior methods to obtain performance improvements (probably, see W4) at an insignificant computational cost. This is demonstrated on NuScenes with TransFusion and DeepInteraction.

**Weaknesses:**

W1) There are a few missing references in related work, for instance
- SpatialDETR from ECCV2022.
- The pioneering work DETR3D from conference on robotics learning.
- TrackFormer, a simple yet elegant approach to multi-object tracking that is completely query-based.

W2) One of the core motivations of the proposed approach is that BEV-based methods propagate information about the /background/ through time, which is unecessary (Section 1 second paragraph and Section 2 last paragraph). This hypothesis is not directly tested. I would have expected an experiment with BEV-based fusion in which the background was removed, to show that this is indeed unecessary.

W3) Section 3.2, which constitutes the core of the proposed approach, is very difficult to understand (or possibly incorrect!). See questions Q1 to Q5 below.

W4) The performance improvements are rather small (0.5 to 0.9 NDS). There is no analysis of standard deviation between different trainings, so it is difficult to appreciate whether these improvements are statistically significant.

W5) There is no analysis of in what scenarios that the proposed approach provide performance improvements. While perhaps not strictly necessary, the performance improvements of the proposed approach are rather small and it is impossible to know whether this is because scenarios where temporal fusion is necessary are scarce in NuScenes or whether the proposed approach does not help so much in such scenarios. A more thorough analysis would be helpful to shed some light on the actual performance of the proposed approach.

Some nitpicks)
- The equations have poor formatting, always starting with a colon and not ending with a comma/period. Moreover, operators are written as variables, e.g., $and$ instead of $\text{and}$.
- $R$ is perhaps not the best choice for the transformation from world-coordinates to ego-coordinates, because $R$ is often used for rotations. In contrast, we also have a translation.
- Equation 2 seems incorrect to me in that it behaves as if $R_{t-1}^t$ was in homogenous coordinates but I suspect that $C_{t-1}$ is not. Moreover, I think it is more common to let $C_{t-1}'\in\mathbb{R}^{3}$ than to put $C_{t-1}'\in\mathbb{R}^{1\times 3}$ in order to have it left of the transformation matrix.
- I think "mathcal" is usually used for sets or operators. It is a bit confusing to see it for the mask and attention matrices.
- It is both clearer and more concise to write $A\in\mathbb{R}^{N_t\times N_{t-1}}$ than "matrix with shape of $N_t\times N_{t-1}$".

**Questions:**

Q1) How is $C_t$ obtained? Is it a fix anchor (like in AnchorDETR), a learnable anchor, or computed from $Q_t$ (like in DETR)?

Q2) What are the details of the transformer architecture? Is it DETR, AnchorDETR, ConditionalDET, DeformableDETR, or something else?

Q3) Is it correct that at each time-step that one would run the model, the MTM is run N+1 times (as shown in Figure 2)? That would not be as efficient as a fully recurrent model. Is this correct, and was a recurrent model considered?

Q4) I do not understand what the cost matrix is, what it is used for, or how it is computed. How is this achieved?

Q5) What is $C_t$? In l130 it seems as if it is a matrix of shape $\mathbb{R}^{K\times 3}$ (or possibly the transpose of that). In equation 3, however, the l2 vector norm is computed, which makes it seem as if $C_t\in\mathbb{R}^3$.

Q6) What is the motivation for going as low as 200 queries? Image-based detectors seem to often adopt 300 queries (e.g., ConditionalDETR) whereas 3D detectors adopt 600 to 1200 (SpatialDETR). BEVFormer adopts 40000 queries, though it uses a sparse attention mechanism.

Q7) What is the standard deviation between experiments? Please remember to not seed dataloading or weight initialization, and to retrain also the TransFusion/DeepInteraction backbone.

Q8) Regarding cross-attention versus MTM qualitative results, could the same be achieved by temperature scaling the cross-attention? Or perhaps by performing cross-attention with positional encodings based on $C_{t-1}'$ instead of $C_{t-1}$ (i.e., correct the position with the predicted velocity of that object)?

Q9) Why is relying on a DETR-like design a limitation ("the main limitation")?

**Limitations:**

The authors provide some discussion of limitations. Though, it is unclear to me why the main limitation is the reliance of a DETR-like design.

---

> ### Author Rebuttal · Authors · 2023-08-09
>
> Thank you for your patient and detailed review. We try to address your comments below.
>
> **Missing references:** Sorry for missing these references. We will cite them into our submitted paper.
>
> **I would have expected an experiment with BEV-based fusion in which the background was removed:** Thanks. We conduct this experiment on BEV-based fusion method MGTANet and then removing all background by the provided ground truth. As shown in table, we surprisingly find that there is a significant performance improvement. However, it is very difficult to thoroughly distinguish foreground and background in practical situations.
>
> |  Method | mAP | NDS |
> |  ----   | :----: | :----: |
> |  MGTANet | 64.0 | 68.1 |
> |  +Remove Background | 94.6 | 82.5 |
>
> **There is no analysis of standard deviation between different trainings:** Thanks. We re-train our method for 3 times as shown in table. The corresponding standard deviation of NDS is $3.3e^{-5}$, which illustrates the stability of our method.
>
> |  # | mAP | NDS |
> |  ----   | :----: | :----: |
> |  1 | 66.47 | 70.86 |
> |  2 | 66.49 | 70.86 |
> |  3 | 66.46 | 70.86 |
>
>
> **There is no analysis of in what scenarios that the proposed approach provide performance improvements:** Valuable suggestion! To illustrate what scenarios that the proposed approach provide performance improvements, we provide the detailed results (mAP) for the categories of Motorcycle and Bicycle (fast-moving small objects), Traffic Cone and Barrier (static objects). We find that our method mainly improves the performance of fast moving small objects. For static objects, our method can maintain a good performance. For the results of more moving objects such as Car and Truck (please refer to the details in our supplemental materials), our method also brings promising performance improvement. We will add this analysis in the final version.
>
> |  Method | Motorcycle | Bicycle | Traffic Cone | Barrier |
> |  ----   | :----: | :----: | :----: | :----: |
> |  TransFusion-L | 71.8 | 56.5 | 74.4 | 71.8 |
> |  +QTNet | 75.5 | 61.5 | 75.4 | 71.4 |
>
>
> **Some errors in equations:** Thanks for pointing out these errors. We will revise them in the final version.
>
>
> ### Some Questions
> **Q1:** The $C_t$ is computed from $Q_t$ by a DETR-like detection head.
>
> **Q2:** Sorry for making you misunderstand. Actually, the mentioned DETR in submitted paper is a DETR-like 3D detector. In particular, we select the TransFusion-L as our default DETR-like detector.
>
> **Q3:** In fact, MTM runs $N$ times rather $N+1$ times. Theoretically, although our method is not more efficient compared with a recurrent model, our MTM is lightweight and only brings a small amount of time overhead compared with the whole network. As shown in table, we can clearly observe that when adopting 5 frames as inputs, it only brings 7.7 ms additional time overhead.
>
> |  Frames | mAP | NDS | FLOPS (G) | Latency (ms) |
> |  ---- | :----: | :----: | :----: | :----: |
> |  1    | 65.0 | 70.0 | 90.65 | 138.2 |
> |  2    | 66.1 | 70.6 | +0.07 | +3.1 |
> |  3    | 66.4 | 70.8 | +0.10 | +4.5 |
> |  4    | 66.5 | 70.9 | +0.14 | +6.5 |
> |  5    | 66.4 | 70.9 | +0.24 | +7.7|
>
> **Q4:** The cost matrix is computed by the L2 distance of objects' center in previous and current frames. It is used to generate the attention map among objects for establishing relationship.
>
> **Q5:**
>
> (i) $C_{t}\in \mathbb{R}^{N\times 3}$ means the centers of $N$ objects.
>
> (ii) Given the current $C_{t}\in \mathbb{R}^{N\times 3}$ and previous $C_{t-1}\in \mathbb{R}^{M\times 3}$, the L2 norms actually means the L2 distance between $C_{t}$ and $C_{t-1}$ so as to obtain the cost matrix $L\in \mathbb{R}^{N\times M}$. We will revise them in final version.
>
> **Q6:** In fact, for a fair comparison, we keep the same value of 200 queries with The LiDAR-based (TransFusion-L) or Multi-modality DETR detectors (e.g. TransFusion, DeepInteraction, BEVFusion). Besides, for highly sparse 3D point cloud, setting 200 queries is usually sufficient to detect most objects
>
> **Q7:**As shown in table, we re-train our methods for 3 times based on TransFusion, and the corresponding standard deviation of NDS is $3.3e^{-5}$. We will make it clear in the final version.
>
> |  # | mAP | NDS |
> |  ----   | :----: | :----: |
> |  1 | 66.47 | 70.86 |
> |  2 | 66.49 | 70.86 |
> |  3 | 66.46 | 70.86 |
>
> **Q8:** Sorry for making you misunderstanding. In fact, we have implemented the cross-attention with the positional encoding based on the correct position (${C^{'}}_{t-1}$) and provide the  corresponding results in Table 5 of the original submitted paper. Here, we further simplify the table and show the results in the following table.
>
> |  # | Cross Attention | MTM | mAP | NDS |
> |  ---- | :----: | :----: | :----: | :----: |
> |  1 |  | | 65.0 | 70.0 |
> |  2 | &#10003; | | 65.2 | 70.0 |
> |  3 |  | &#10003; | 66.2 | 70.5 |
>
> **Q9:** To our best knowledge, DETR-like approaches are still developing in the 3D domain. In this paper, we mainly verify the superiority of our method for DETR-like 3D detectors. The effectiveness of 3D detectors for other paradigms remains to be explored in future research. We will make further clarification on limitations.

---

> > ### Comment · Reviewer_LNvD · 2023-08-11
> >
> > Thank you for the thorough rebuttal. I would have two follow-up questions
> >
> > - Regarding W2: It is not sound to remove the ground-truth from the propagated feature maps, because one then essentially feeds the ground-truth (or something highly correlated with it) into the model. I would instead suggest to use the model's prediction to remove the parts of the BEV feature map based on the predictions that the model makes.
> >
> > - Regarding W4: How is the standard deviation computed? Do the neural networks use different weight initializations? What about the backbone? How about the dataset shuffling? Is anything seeded? I find the reported number to be substantially lower than what I usually encounter on NuScenes.

---

> > > ### Author Response · Authors · 2023-08-13
> > >
> > > Thanks for your comments.
> > >
> > > **W2:** Thanks for your suggestion. We use the prediction to remove the background of BEV features. As shown in table, removing the background does not downgrade the performance.
> > >
> > > | Method | mAP | NDS |
> > > | --- | :---: | :---:|
> > > | MGTANet | 64.0 | 68.1 |
> > > | +Remove Background | 64.1 | 68.2 |
> > >
> > > **W4:** Sorry for making you misunderstanding. The main reason for our stable results (low  standard deviation) is that we adopt different training strategies with some methods in NuScenes. Actually, we have described our training procedure in the implementation details of the main paper, which includes a two-stage training manner. In more detail, we load the weights of the backbones from the trained TransFusion/DeepInteraction and freeze their parameters in this first stage, which can promise consistent detection performance for a fair comparison. In the second stage, we focus on applying our proposed temporal fusion to refine the detection result in the first stage. In a word, the uncertainty of our detection performance is only from the second stage. Thus, we run our model three times, as shown in table , and find the final performance is very stable after fixing the first stage. We will make it more clear in the final version.
> > >
> > > | # | mAP | NDS |
> > > | --- | :---: | :---:|
> > > | 1 | 66.47 | 70.86 |
> > > | 2 | 66.49 | 70.86 |
> > > | 3 | 66.46 | 70.86 |

---

> > > > ### Comment · Reviewer_LNvD · 2023-08-14
> > > >
> > > > Thanks for the replies and the swift provision of experiments. My updated assessment is the following:
> > > >
> > > > - **W4** Performance improvements are rather small, but for some classes the proposed approach seems helpful.
> > > >
> > > > - **W3** The notation in the method section is confusing. The rebuttal does provide some answers, but it is no guarantee that those answers will be integrated into the text into a manner that is easy to understand.
> > > >
> > > > I will update my rating from 3 to 4.

---

> > > > > ### Author Response · Authors · 2023-08-15
> > > > >
> > > > > Thank you for your reply and decision to raise score.
> > > > >
> > > > > **W4:** Thanks for your comments. On the one hand, we compare our method with other methods on the same baseline. As shown in table, our method brings more improvements with negligible computation and runtime cost, which is very important for autonomous driving.
> > > > >
> > > > > |  Method | mAP | NDS | FLOPs (G) | Latency (ms) |
> > > > > |  ----   |  :----:  | :----: | :----: | :----: |
> > > > > |  baseline | 63.1 | 67.8 | 90.7 | 138.2 |
> > > > > |  +MPPNet | 63.4 | 68.2 | +131.3 | +127.9 |
> > > > > |  +MGTANet| 64.0 | 68.1 | +193.0 | +22.1 |
> > > > > |  +QTNet | 64.7 | 69.0 | +0.1 | +4.5 |
> > > > >
> > > > >  On the other hand, it is difficult to improve the performance on the advanced baseline (e.g. DeepInteraction). As shown in table, our method brings 0.5 NDS improvement on the DeepInteraction baseline, which is the same as the concurrent work FocalFormer (ICCV 2023). In other words, the improvement is acceptable.
> > > > >
> > > > > |  Method | Year | mAP | NDS |
> > > > > |  ----   | ---- | :----: | :----: |
> > > > > |  DeepInteraction | NeurIPS 2022 | 69.9 | 72.6 |
> > > > > |  FocalFormer | ICCV 2023 | 70.5 | 73.1 |
> > > > > |  QTNet | - |  70.3 | 73.1 |
> > > > >
> > > > > Besides, we validate our method on the camera-based method (CAPE CVPR 2023) in the supplement materials. There is a significant performance improvement.
> > > > >
> > > > > **W3:** Thanks for your comments. We will revise these confusing notations by a more comprehensible manner. Specifically, We will make these notation more clear by following your suggestions, correct these mistakes, and integrate these answers in rebuttal into our revised paper.

---

### Official Review · Reviewer_DMZS · 2023-07-03

**Soundness:** 4 excellent
**Presentation:** 4 excellent
**Contribution:** 3 good
**Rating:** 6
**Confidence:** 5

**Summary:**

The paper introduced a Query-based Temporal Fusion Network. It faciliate the object queries in previous frame to enhance the
current object queries by aproposed Motion-guided Temporal Modeling module, which utilizes both spatial and motion information to construct a cost matrix for efficient temporal fusion. The proposed method significantly improve upon the query-based baseline while incurring a negligible runtime cost.

**Strengths:**

1. The paper introduces a novel temporal fusion paradigm that directly utilizes query-based features for achieving temporal fusion.
2. The proposed framework significantly improves upon the query-based baseline method while incurring a negligible runtime cost.
3. The paper is well-written, and the clarity of the figure enhances the understanding of the proposed concepts.

**Weaknesses:**

1. The novelty of this work is limited. This work is built on top of existing DETR-based detector and perform temporal fusion on query-based features. The idea is quite similar to CenterFormer. What makes different is that the query-based features are generated from DETR detector instead of sampling from heat map in CenterFormer. The memory bank is also not new.
2. As the method use TransFusion as backbone model, it would be worth exploring the use of intermediate BEV features as the key and value for temporal fusion.  These features are expected to offer a richer context and can be also cached in the memory bank.
3. Using the velocity \times time to align the features across frames is not able to generalize to long-sequence as the object motion is not always constant and the turning is not considered.


**Questions:**

It is better to publish the results on NuScenes leaderboard and validate its method on Waymo Open Dataset.

**Limitations:**

The author addressed the limitations.

---

> ### Author Rebuttal · Authors · 2023-08-09
>
> Thank you for your patient and detailed review. We try to address your comments below.
>
> **Difference from CenterFormer:** Thanks. In fact, our method is different from CenterFormer. CenterFormer conducts the temporal fusion between the current queries and historical BEV features, which is a sparse-to-dense strategy. But our method conducts the temporal fusion between current and historical queries, which is a sparse-to-sparse strategy. For the memory bank, we do not regard it as our contribution since it is only used to store historical information for avoiding redundant computation.
>
> **It would be worth exploring the use of intermediate BEV features as the key and value for temporal fusion:** Thanks. As shown in Table, we conduct the temporal fusion by achieving the interaction between sparse object queries (as Q) and dense BEV features (as K or V). We find that our method brings more improvement than directly conducting the temporal fusion between queries and BEV features. We think that establishing the relationship between current queries and historical BEV features is difficult since there are many backgrounds in BEV features, which hinders the instance-level temporal fusion. Besides, object queries have already aggregated context information from BEV features. Therefore, we think our proposed sparse-to-sparse temporal fusion is reasonable.
>
> |  Method | Q | K/V | mAP | NDS |
> |  ----   | :----: | :----: | :----: | :----: |
> |  TransFusion-L | - | - | 65.0 | 70.0 |
> |  +QTNet | Query Features | BEV Features | 65.3 | 70.1 |
> |  +QTNet | Query Features | Query Features | 66.2 | 70.6 |
>
> **Using motion to align the features across frames is not able to generalize to long-sequence as the object motion is not always constant and the turning is not considered:** Actually, our method achieves temporal fusion between two adjacent frames in a progressive temporal fusion mechanism (Please refer to the structure figure in the uploaded pdf of our rebuttal). Besides, we compute the mean error of velocity (mAVE) between adjacent frames as $0.24 m/s$, which is relatively small. Thus, we argue that our method is applicable to the long-sequence cases.
>
> **Publish the results on NuScenes leaderboard and validate on Waymo:** Thanks for your valuable suggestion. We publish the results on NuScenes leaderboard, which is included in the uploaded pdf. Besides, as shown in table, we integrate our QTNet into ConQueR on the Waymo dataset. Due to the limited time and computation resources, we only train all models on the 20% sequences (keep temporal consistency) of training set and validate on the full validation set. We find that there is an obvious performance improvement, illustrating the effectiveness of our method.
>
> |  Method | Veh | Ped | Cyc | L2 mAPH |
> |  ----   | :----: | :----: | :----: | :----: |
> |  ConQueR | 58.7 | 62.7 | 49.3 | 56.9 |
> |  +QTNet | 59.5 | 63.3 | 54.1 | 59.0 |

---

> > ### Comment · Reviewer_DMZS · 2023-08-20
> >
> > After carefully reading other comments, I believe the solution proposed by the author is effective, but lacks references and discussion of related work, and the innovation is limited. I will maintain my rating.

---

### Official Review · Reviewer_BRpP · 2023-07-06

**Soundness:** 2 fair
**Presentation:** 3 good
**Contribution:** 2 fair
**Rating:** 4
**Confidence:** 4

**Summary:**

This paper proposes a new strategy of fusing temporal information for camera-lidar based 3D object detectors. The main method is to design a plug-and-play module, which uses predicted velocity and vehicle ego information to compute the correspondence matrix among queries explicitly.  This module can be integrated into some advanced LiDAR-only or multi-modality 3D detectors.


**Strengths:**

1. The paper provides a clear and well-structured overview of the approach, making it easy for readers to understand and follow.
2. The proposed method brings competitive performance with negligible computation cost and latency on the nuScenes dataset.

**Weaknesses:**

1. The experiments are not very sufficient. For detectors of the DETR architecture, I think using explicit geometric constraint to match queries may not be necessary. Maybe directly conducting Attention operation among current queries and historical queries are enough to bring promising performance, just like StreamPETR. However, the authors do not analyze this issue.
2. The method proposed in this paper does not bring enough benefits, This method works on the nuScenes dataset. It may be that  nuScenes has a large proportion of static data. I hope to verify it on other data sets. That will affect whether I am willing to improve the score.

**Questions:**

1. Could you please provide your baseline(DeepInteraction or Transfusion-L) multi-frame results, just aligning the previous frame to the current frame? I hope to know how much gain is brought by MTM, rather than multi-frame alignment enhancing feature representation.
2. Could you please provide some visual samples? MTM should alleviate the problem of direction misdetection.

**Limitations:**

As mentioned before, using explicit geometric constraint violates the simplicity of DETR. Thus, I do not think the proposed strategy will be widely adopted by future work, although the Query-based alignment strategy shows benefits compared with the BEV-based and Proposal-based methods. Anyway, I hope the authors can provide me a discussion comparing the proposed Query-based strategy and directly conducting Attention among current and historical queries.

---

> ### Author Rebuttal · Authors · 2023-08-09
>
> Thank you for your patient and detailed review. We try to address your comments below.
>
> **Analyze the attention and MTM:** Good suggestion. As shown in table, We replace the MTM with attention operation and find that there is no noticeable performance gain. The main reason is that objects with similar geometric structures between adjacent frames are difficult to distinguish in LiDAR domain, which makes the association between two frames less reliable. However, StreamPETR can work well by building the relationship of queries between two frames in camera domain due to the rich appearance information in 2D images.
>
> |  Method | mAP | NDS |
> |  ----   | :----: | :----: |
> |  TransFusion-L | 65.0 | 70.0 |
> |  +StreamPETR | 65.2 | 70.0 |
> |  +MTM | 66.2 | 70.5 |
>
> **The method proposed in this paper does not bring enough benefits:** Thanks. We provide the results of our MTM on the Waymo dataset in table. Here, we select the representative DETR-like method ConQueR (CVPR 2023) as our baseline. Due to the limited time and computation resources, we only train all models on  training set with only 20% sequences to keep temporal consistency. As shown, our method even brings more obvious performance improvement on Waymo dataset, which illustrates its effectiveness.
>
> |  Method | Veh | Ped | Cyc | L2 mAPH |
> |  ----   | :----: | :----: | :----: | :----: |
> |  ConQueR | 58.7 | 62.7 | 49.3 | 56.9 |
> |  +QTNet | 59.5 | 63.3 | 54.1 | 59.0 |
>
> **Provide your baseline multi-frame results, just aligning the previous frame to the current frame:** Thanks. We align the previous queries to the current frame and send them to the decoder for comparison. As shown in table, directly transfering the past queries to current frame only brings 0.1% mAP improvement. In contrast, our MTM brings more promising performance improvement with 1.2% mAP.
>
> |  Method | mAP | NDS |
> |  ----   | :----: | :----: |
> |  TransFusion-L | 65.0 | 70.0 |
> |  +Propagate | 65.1 | 70.0 |
> |  +MTM | 66.2 | 70.5 |
>
> **Provide some visual samples:** Thanks. In fact, we have provided the visualization of detection results in the supplement materials. Besides, following your valuable suggestion, we highlight some visual samples for direction misdetection in the uploaded pdf of our rebuttal.

---

### Official Review · Reviewer_ycEE · 2023-07-08

**Soundness:** 2 fair
**Presentation:** 2 fair
**Contribution:** 2 fair
**Rating:** 5
**Confidence:** 5

**Summary:**

In this paper, the authors propose a simple and effective Query-based Temporal Fusion Network (QTNet). The main idea is to exploit the object queries in previous frames to enhance the current object queries by the proposed Motion-guided Temporal Modeling (MTM)  module,  Experimental results show the proposed QTNet outperforms BEV-based or proposal-based manners on the nuScenes dataset.

**Strengths:**

1. The proposed QTNet can be plugged into LiDAR-only or multi-modality 3D detectors.
2. QTNet can boost 3D detector's performance with negligible computation cost and latency.


**Weaknesses:**

1) The innovation of the method is limited. The fusion of temporal features using motion information has similar ideas in both the proposal-based method （MPPNet，MSF [1]）and the query-based method （MOTRV2 [2]）. For example, MOTRV2 greatly improves the positioning and tracking performance of the 2D transformer-based tracker by fusing the timing query features and passing the location of the timing query.
2) The comparison with proposal-based methods is not comprehensive.

      a) The fusion strategy of MPPNet only uses the historical proposal information that can match the current moment, that is, forms the trajectory, and discards other frames. This means that MPPnet cannot use historical information to restore frames missed at the current moment. The author can use motion information to transfer the unmatched past proposals to the current frame, like query-based startegy, and then verify the performance of MPPNet in this way, which can more fairly verify the performance of query-based strategy and proposal-based strategy.

     b) The comparison with SOTA methods, such as MSF, is missing. MSF also uses the motion-guided feature fusion strategy and achieve higher performance and efficiency than MPPNet.

3. The ablation experiments of MTM are not sufficient enough to verify the superiority of MTM design. The author should provide a simple baseline that just transfer all motion-aligned past queries to the current frame with an NMS to remove redundant temporal queries, and then send them to the decoder  with the current query together. Compared with this baseline, we can know more clearly whether it is the gain brought by the MTM attention design or just temporal queries.

[1] MSF: Motion-guided Sequential Fusion for Efficient 3D Object Detection from Point Cloud Sequences
[2] MOTRv2: Bootstrapping End-to-End Multi-Object Tracking by Pretrained Object Detectors



**Questions:**

See Weakness.

**Limitations:**

See Weakness.

---

> ### Author Rebuttal · Authors · 2023-08-09
>
> Thank you for your patient and detailed review. We try to address your comments below.
>
> **Difference of using motion information with MPPNet and MSF:** Thanks. The main idea of our QTNet is different from MPPNet and MSF. MPPNet utilizes the velocity prediction for motion compensation in tracking, and then generate trajectory for temporal fusion. MSF utilizes the motion for propagating the current detection boxes to history frames for sampling point clouds. However, our method utilize the motion to establish the attention map among queries and fuse these queries by transformer, which is a more efficient way for temporal fusion.
>
> **Difference from MOTRv2:** MOTRv2 propagates the historical queries as the track queries in the current frame. Then it concatenates these queries as the $Q$ of transformer decoder, and the dense image features as the $K$ and $V$ of transformer decoder. In other words, MOTRv2 is a sparse-to-dense fusion strategy while our proposed MTM conducts the temporal fusion among sparse queries, which can be regarded as a sparse-to-sparse fusion strategy. Besides, MOTRv2 can work well by building the relationship in camera domain due to the rich appearance information in 2D images. However, objects with similar geometric structures between adjacent frames are difficult to distinguish in LiDAR domain, which makes the association between two frames less reliable. Therefore, we design the MTM operation to solve this problem.
>
> **Comparison with proposal-based methods:** Thanks. We propagate the historical detection results to the current frame to verify the performance MPPNet. As shown in table, there is a large performance degradation. The behind reason is that propagation operation may leads to a lot of false positive predictions and harm the final prediction performance.
>
> |  Method | Propagate | mAP | NDS |
> |  ----   |  :----:  | :----: | :----: |
> |  baseline |  | 63.1 | 67.8 |
> |  +MPPNet |  | 63.4 | 68.2 |
> |  +MPPNet | &#10003; | 61.5 | 67.3 |
>
> Besides, as shown in table, we compare MSF with MPPNet on the same baseline on the nuScenes dataset. Although MSF has lower latency than MPPNet, MSF produces worse performance. The main reason is that MSF only utilizes the velocity to move current boxes to previous frames and does not take into account the turning angle, which limits MSF generalize to long-sequence.
>
> |  Method | mAP | NDS | FLOPs (G) | Latency (ms) |
> |  ----   |  :----:  | :----: | :----: | :----: |
> |  baseline | 63.1 | 67.8 | 90.7 | 138.2 |
> |  +MPPNet | 63.4 | 68.2 | +131.3 | +127.9 |
> |  +MSF | 62.9 | 67.9 | +198.3 | +81.5 |
> |  +QTNet | 64.7 | 69.0 | +0.1 | +4.5 |
>
> **Comparison with simple baseline that just transfer all motion-aligned past queries to the current frame:** Thanks. We transfer the past queries to the current frame and send them to the decoder ('+Propagate' in table). We find that this temporal fusion manner does not bring obvious performance improvement. The main reason is that there lacks obvious feature distinction in LiDAR domain, which leads to having a difficulty in learning attention among object queries. However, MTM makes the learning process easier by our proposed explicit geometric constraint.
>
> |  Method | mAP | NDS |
> |  ----   | :----: | :----: |
> |  TransFusion-L | 65.0 | 70.0 |
> |  +Propagate | 65.1 | 70.0 |
> |  +MTM | 66.2 | 70.5 |

---

> > ### Comment · Reviewer_ycEE · 2023-08-21
> >
> > Thank the authors for the response and additional experiments. After the rebuttal my concerns are resolved and I keep my my original rating.

---

### Official Review · Reviewer_T8P4 · 2023-07-08

**Soundness:** 3 good
**Presentation:** 3 good
**Contribution:** 1 poor
**Rating:** 4
**Confidence:** 4

**Summary:**

This paper introduces an approach that leverages object queries as a form of temporal memory. By establishing an attention map between queries from current and previous frames, the method effectively captures and encodes explicit motion information using velocity prediction and pose transformation. The proposed approach, referred to as QTNet, achieves superior performance compared to BEV-based or proposal-based techniques when evaluated on the nuScenes dataset. Furthermore, the Motion-guided Temporal Modeling Module (MTM) module can be seamlessly integrated as a plug-in component into other existing methods.

**Strengths:**

1. This article is well-written and can be easily reproduced based on its content.
2. The visualization of the attention map reveals the reasons behind the superior performance of the proposed method.
3. Despite its simplicity, this approach achieves significant performance improvements.
4. Decouple Strategy is interesting.

**Weaknesses:**

1. In line 126, the memory bank stores $Q_{t - 1}$, however in line 154, you actually use previous fused queries $Q_{t - 1}'$.
2. This article lacks innovation as it employs a manually defined attention map based on simple projections of velocity and pose, and the memory bank consists of straightforward query storage. Additionally, the performance of this approach is reliant on velocity prediction and tends to degrade in congested scenes.
3. There should be more exploration and comparison of different paradigms. Currently, each paradigm is represented by only one method, lacking universality. The author could validate their approach using a simpler paradigm, such as employing the straightforward BEV feature query method used in BEVfusion. This kind of comparison would be more reliable and credible.

**Questions:**

Please see the Weaknesses

---

> ### Author Rebuttal · Authors · 2023-08-09
>
> Thank you for your patient and detailed review. We try to address your comments below.
> ### W1: In line 126, the memory bank stores $Q_{t-1}$ , however in line 154, you actually use previous fused queries ${Q^{'}}_{t-1}$.
> Sorry for the misunderstanding. In fact, we only store historical queries $[Q_{t-1}, Q_{t-2}, ...Q_{t-N}]$ in the memory bank and then fused these historical queries to generate the ${Q^{'}}_{t-1}$ for enhancing current queries $Q_t$. We will make it clear in this original paper.
>
> ### W2: Manually defined attention map based on simple projections of velocity and pose, and the performance of this approach is reliant on velocity prediction and tends to degrade in congested scenes.
> Thanks. We summary your concerns as follows:
>
> **About manually defined attention map:** The defined attention map is simple, effective, and well-designed. Actually, 3D objects obey the physical law of motion in the real-world 3D space (LiDAR domain), which means that the same 3D object between adjacent frames does not shift too much in a short time. Therefore, utilizing this prior information is important to define the attention map among object queries, which makes our temporal model know where to pay attention, resulting in better performance.
>
> **About the reliance on velocity prediction:** To achieve promising detection performance, velocity prediction of objects is the critical information for almost all temporal-based 3D detectors (e.g. MPPNet, MSF, MGTANet) rather than only our method.
>
> **About the performance on congested scenes:** Since nuScence dataset does not provide the settings in the congested scene, we simply divide it into two scenes, which contains the crowded scenes (the number of objects is greater than 80) and the uncrowded scenes (the number of objects is less than 80). The corresponding results are shown in table. We can observe that our approach can obtain more obvious performance improvement in crowded scenes than that of uncrowded scenes, which illustrates the effectiveness of our proposed method.
>
> |  Method | Scenes | mAP | NDS  |
> |  ----   | :----: | :----: | :----: |
> |  TransFusion-L | uncrowded | 67.0 | 69.9 |
> |  QTNet | uncrowded | 67.9 | 70.7  |
> |  TransFusion-L | crowded | 64.4 | 68.7 |
> |  QTNet | crowded | 65.8 | 69.7 |
>
> ### W3: There should be more exploration and comparison of different paradigms.
>
> **More exploration and comparison of different paradigms:** Thanks. The BEV-based paradigm conducts the temporal fusion on the dense BEV features, which may bring a lot of unnecessary computation cost on background. The proposal-based paradigm needs time-consuming operations to generate sparse 3D proposal features, and the performance highly depends on the quality of 3D proposals. In contrast, our query-based paradigm is based a sparse feature representation and can effectively aggregate the foreground object information, which is effective and efficient. Besides, our method can get rid of complex 3D RoI operations and less sensitive to 3D object of size and orientation than proposal-based representation.
>
> **More models for comparison:** Actually, we have selected one of the advanced and representative temporal fusion methods for each paradigm as comparison. To further illustrate the universality of our method, we select a new proposal-based method MSF, an improved version for MPPNet, as comparison. We can observe that QTNet consistently outperforms MSF in terms of performance and efficiency.
>
> |  Method | mAP | NDS | FLOPs (G) | Latency (ms) |
> |  ----   | :----: | :----: | :----: | :----: |
> |  baseline | 63.1 | 67.8 | 90.7 | 138.2 |
> |  +MPPNet | 63.4 | 68.2 | +131.3 | +127.9 |
> |  +MSF | 62.9 | 67.9 | +198.3 | +81.5 |
> |  +QTNet | 64.7 | 69.0 | +0.1 | +4.5 |
>
> **Validate on the BEVFusion:** We integrate our temporal fusion into BEVFusion by utilizing our method on the queries of BEVFusion. The experimental results show QTNet brings further performance improvement to BEVFusion with a small latency.
>
> |  Method | Modality | mAP | NDS | Latency (ms) |
> |  ----   | :----: | :----: | :----: | :----: |
> |  BEVFusion | LC | 69.6 | 72.1 | 965.5 |
> |  +QTNet | LC | 70.1 | 72.5 | +6.5 |

---

### Author Rebuttal · Authors · 2023-08-09

We upload a pdf, which contains the visualization about orientation, results on the nuScenes leaderboard, and the illustration about our progressive temporal fusion mechanism.

---

### Decision · Program_Chairs · 2023-09-21

**Decision:**

Accept (poster)

**Comment:**

The paper received mixed reviews. ycEE requests more comparisons with proposal-based methods and more ablation studies, which are adequately provided in the author rebuttal. T8P4 also requests more comparisons, which are provided in the rebuttal and authors are encouraged to include them in the revised version. BRpP questions the significance of the benefits and further analysis of the method, which are provided in the rebuttal and indicated by the reviewer as sufficient. DMZS states similarity to CenterFormer, for which the author response satisfactorily outlines the distinction and the reviewer maintains a rating to accept. LNvD requests additional analysis of standard deviations and scenarios that lead to improvements, which are addressed by the rebuttal and which the authors may include in the revised version. Overall, the meta-reviewer agrees that the concerns raised by the reviews have been comprehensively addressed by the rebuttal and the paper may be accepted.